# Integrating microbial abundance time series with fermentation dynamics of the rumen microbiome *via* mathematical modelling

**Mohsen Davoudkhani**[1⊙], **Francesco Rubino**[2⊙], **Christopher J. Creevey**[2], **Seppo Ahvenjärvi**[3], **Ali R. Bayat**[3], **Ilma Tapio**[4], **Alejandro Belanche**[5], **Rafael Muñoz-Tamayo**[1]*

**1** INRAE, AgroParisTech, UMR Modélisation Systémique Appliquée aux Ruminants, Université Paris-Saclay, Palaiseau, France, **2** Institute of Global Food Security, School of Biological Sciences, Queen's University Belfast, Northern Ireland, United Kingdom, **3** Animal Nutrition, Production Systems, Natural Resources Institute Finland (Luke), Jokioinen, Finland, **4** Genomics and Breeding, Production Systems, Natural Resources Institute Finland (Luke), Jokioinen, Finland, **5** Departamento de Producción Animal y Ciencia de los Alimentos, Universidad de Zaragoza, Zaragoza, Spain

⊙ These authors contributed equally to this work.
* Rafael.munoz-tamayo@inrae.fr

**Data Availability Statement:** The scripts for the inference of the functional modules are available at https://github.com/frubino/cowpi and https://doi.

## Abstract

The rumen represents a dynamic microbial ecosystem where fermentation metabolites and microbial concentrations change over time in response to dietary changes. The integration of microbial genomic knowledge and dynamic modelling can enhance our system-level understanding of rumen ecosystem's function. However, such an integration between dynamic models and rumen microbiota data is lacking. The objective of this work was to integrate rumen microbiota time series determined by 16S rRNA gene amplicon sequencing into a dynamic modelling framework to link microbial data to the dynamics of the volatile fatty acids (VFA) production during fermentation. For that, we used the theory of state observers to develop a model that estimates the dynamics of VFA from the data of microbial functional proxies associated with the specific production of each VFA. We determined the microbial proxies using CowPi to infer the functional potential of the rumen microbiota and extrapolate their functional modules from KEGG (Kyoto Encyclopedia of Genes and Genomes). The approach was challenged using data from an *in vitro* RUSITEC experiment and from an *in vivo* experiment with four cows. The model performance was evaluated by the coefficient of variation of the root mean square error (CRMSE). For the *in vitro* case study, the mean CVRMSE were 9.8% for acetate, 14% for butyrate and 14.5% for propionate. For the *in vivo* case study, the mean CVRMSE were 16.4% for acetate, 15.8% for butyrate and 19.8% for propionate. The mean CVRMSE for the VFA molar fractions were 3.1% for acetate, 3.8% for butyrate and 8.9% for propionate. Ours results show the promising application of state observers integrated with microbiota time series data for predicting rumen microbial metabolism.

org/10.5281/zenodo.8401851. The data and scripts
of the implementation of the observer for each
case study are available at https://doi.org/10.5281/
zenodo.8386786 [48]. Sequencing data for the in
vitro case study are accessible at the EBI Short
Read Archive from the European Nucleotide
Archive (accession number PRJEB20255). The
sequencing data for the in vivo case study are
accessible at NCBI SRA under the BioProject
PRJNA1023082.

**Funding:** All authors receive funding from the
MASTER project, an Innovation Action funded by
the European Union's Horizon 2020 research and
innovation programme under grant agreement No
818368. The funders had no role in study design,
data collection and analysis, decision to publish, or
preparation of the manuscript.

**Competing interests:** The authors have declared
that no competing interests exist.

## Introduction

The function of the rumen microbiota affects animal production phenotypes, feed efficiency
and methane emissions [1]. Our knowledge on the structure and function of the rumen micro-
biota has been greatly improved due to the progress on culture independent omic techniques
[2, 3]. A powerful use of these techniques is the analysis of microbial time series data allowing
to characterise the dynamics of the rumen microbial ecosystem. Applications include the
study of microbial colonization of feed particles [4–6], microbial resilience in response to per-
turbations [7], impact of acidotic challenge on microbial function [8], activity of methanogens
in response to the supplementation of a methane inhibitor [9] and evolution of gut microbiota
through weaning transition [10, 11].

In parallel to the use of omics techniques to characterise rumen microbiota patterns,
dynamic models have been developed to represent the rumen fermentation profile under *in
vitro* [12, 13] and *in vivo* [14–16] conditions. Existing models of rumen function consider an
aggregated representation of the rumen microbiota and its metabolic function. However, none
of these models integrate microbial genomic knowledge and thus do not capitalize on the rich
information that microbial genomic sequencing provides. Integration of dynamic modelling
and microbial data has the potential to improve the understanding of the rumen ecosystem, to
enhance predictive power of rumen models and to help the design of microbial manipulation
strategies to improve rumen function [17]. Recently, some studies have applied the genome-
scale metabolic approach to reconstruct metabolic networks of rumen microbes species [18–
20] and to predict the metabolism of minimal rumen microbial consortium [21]. Another
model approach consists in exploiting microbial time series data, with a variety of dedicated
mathematical approaches [22] including the generalized Lotka-Volterra (gLV) model [23, 24].
In its standard form, the gLV approach determines interactions between microbes but it does
not provide either mechanistic insights or predictions on the dynamics of microbial metabo-
lism. An attempt to couple the gLV approach with a kinetic metabolic model that integrates
both microbial interactions and metabolism in a nitrification reactor was developed in [25].
The model was useful to identify qualitative aspects of the system such as the coexistence of
two competing bacteria. However, this type of modelling approach based on pairwise micro-
bial interactions is limited to ecosystems with few species due to the high number of parame-
ters that need to be estimated.

In this work, we aim at integrating microbial time series within a dynamic modelling
framework *via* the application of state observers (also called software sensors). An observer is
an algorithm that combines measurements and a mathematical model to estimate unmeasured
variables (see, *e.g.*, [26] for a review). Within the classes of state observers, asymptotic observ-
ers are of particular interest since they do not require knowledge on the kinetic functions rep-
resenting the reaction rates. Asymptotic observers have been applied in simple microbial
processes with few microbial species or in processes where the microbiota is represented in
aggregated fashion by few microbial functional groups [27, 28]. Observers in combination
with optimization routines have been used to assign the metabolic functions of species (Opera-
tional Taxonomic Units—OTUs) participating in a nitrification process [29, 30]. In these pre-
vious works, the functional assignment of species was translated into an optimization problem
which assumes that one species participates only in one reaction. Such a hypothesis might hold
for the nitrification process. However, the assumption of single microbial function does not
seem to apply to the rumen microbiota, since microbes participate in various metabolic path-
ways simultaneously. To circumvent the single functional constraint, here we propose a novel
approach that uses OTU data to derive microbial functional proxies for specific process of
rumen metabolism such as volatile fatty acid (VFA) production. Our approach was tested

firstly using published experimental data from an *in vitro* study [5]. We presented preliminary results of this approach at the 10th Workshop on Modelling Nutrient Digestion and Utilization in Farm Animals (MODNUT, 18th–21st September 2022, Sardinia, Italy). The corresponding abstract is published in the conference proceedings [31]. We performed further an experiment with four cows to assess the modelling approach under *in vivo* conditions. Following open science practices [32], the data and scripts are freely available.

## Material and methods

In this section, we provide a general overview of the experimental case studies. Next, we describe the modelling approach.

### *In vitro* case study

We used data from an *in vitro* experiment carried out in RUSITEC (Rumen Simulation Technique) systems [5] to study the feed microbial colonization dynamics of fresh ryegrass or ryegrass hay. Rumen inoculum was obtained from rumen fistulated cows, and the incubation ran for 18 consecutive days in 16 vessels. Fermentation vessels (800 mL effective volume) were inoculated with rumen fluid and incubated with the experimental diets (80:20 forage to concentrate ratio). One bag containing feed (11.25 g DM) was daily supplied to each vessel and incubated for 48h. Artificial saliva was continuously infused at a dilution rate of 3.35%/h (equivalent to 645 mL/d). After 14 days of adaptation to the experimental diets, the fermentation dynamics was monitored during 3 consecutive days. For each day, samples were taken at 2, 4, 8 and 24 h for determination of acetate, butyrate and propionate and for microbial characterization. Metabolite concentrations for each sampling time was reported as the mean value of the three samples. For microbial characterization, the samples were pooled per time point. The microbial RNA was extracted from the liquid and solid phases and the bacterial community structure was characterized by 16SrRNA (cDNA) Next Generation Sequencing (NGS) as previously described [5]. Raw sequence reads are accessible at the EBI Short Read Archive from the European Nucleotide Archive (accession number PRJEB20255).

### *In vivo* experiment

We conducted an experiment with four Nordic Red dairy cows, selected based on similar calving dates and equipped with rumen fistulas to provide dynamic data to assess our modelling approach *in vivo*. For 10 days of diet adaptation period, cows were offered total mixed ration consisting of grass silage (timothy-meadow fescue sward) preserved with formic acid based additive (AIV Ässä Na; 5 litres/tonne) provided at 45:55 forage to concentrate ratio on a dry matter basis. Concentrate mixture consisted of barley 210, oats 210, wheat 100, sugar beet pulp 220, rapeseed meal 230, and a mixture of minerals and vitamins 30 g/kg on as fed basis. Following the adaptation period, cows were located to metabolic chambers. Day 1 was dedicated to adaptation to the chamber conditions, while on d 2 and d 3 gas exchanges were measured for 48 h (for details see [33]). To determine circadian changes in rumen fermentation and rumen microbial community composition, on chamber d 4 and d 5 for 48 h period rumen liquid samples were collected every 3 h through ruminal fistula from the ventral site of the rumen using 500-mL bottle. Two sub-samples were taken for VFA and ammonia-N determination as described in [11] and were stored at –20°C for later analysis. Rumen liquid samples for microbial analysis were mixed to get homogenous distribution of microorganisms in the liquid, aliquoted into 2-mL sterile screw cap tubes, snap frozen on dry ice and stored at -80°C until DNA extraction. The total DNA was extracted from 500 μL of rumen liquid following protocol described in [34]. Libraries of the bacterial 16S ribosomal RNA (rRNA) V4 region were

prepared using 515F and 806R primers [35] and sequenced on Illumina MiSeq (Finnish Functional Genomics Centre, Turku) using the Paired-End approach and 2 x 250 bp chemistry. The sequencing data are accessible at NCBI SRA under the BioProject PRJNA1023082.

The cow experiments were conducted at the Luke Minkiö research dairy barn (Finland) and all experimental procedures were approved by the Project Authorisation Board (Regional Administrative Agency for Southern Finland, Hämeenlinna, Finland; ESAVI/34265/2019) in accordance with the guidelines established by the European Community Council Directive 86/609/EEC.

### Asymptotic observer and microbial functional proxies

Let us consider a metabolic process occurring in a continuous reactor where $n$ microbial species $x_i$ grow at specific kinetic rates $r_i$ and produce the compound $s$. We consider that the reactor has a known dilution rate $D$ and that the microbes leave the system at a lower rate than the dilution rate, which implies that the residence time of microbes is higher than the residence time of soluble compounds. The higher microbial residence time results from biological phenomena such as attachment to solid particles and formation of microbial aggregates. A simple way to model this phenomena is by adding a residence time factor $\alpha$ [36]. The dynamics of $s$ and a microbial species $x_i$ follow:

$$\frac{dx_i}{dt} = r_i - \alpha D x_i$$

$$\frac{ds}{dt} = \sum_{i=1}^{n} Y_{s,i} r_i - Ds$$

with $Y_{s,i}$ the production yield for the reaction $r_i$. The kinetic rates $r_i$ can be represented by a mathematical function like the Monod or Haldane equations. These kinetic functions are defined by specific parameters for each microbe $x_i$. The model equations [1, 2] assume that the microbe $x_i$ participates only in reaction $r_i$ which does not hold for many rumen microbes. We developed here an alternative approach adapted to the rumen microbiota that overpasses the need of assigning functionalities to microbial species. As observed in equation [2], the production of $s$ results from the collective metabolic activity of the microbial consortium. In our approach, we assumed that the production of $s$ can be related to a subunit of the specific metabolic pathways involved in the production of $s$. This subunit gathers the functional activity of the full consortium and can be used as a proxy of microbial activity of the rumen microbiome. The construction of the microbial proxy will be detailed later on. We will call $m_j$ the microbial proxy associated with the production of $s_j$. Here, our compounds $s_j$ are the major VFA from rumen fermentation: acetate ($s_{ac}$), butyrate ($s_{bu}$) and propionate ($s_{pr}$).

For the compound $s_j$, the resulting model is defined by two equations

$$\frac{dm_j}{dt} = \rho_j - \alpha D m_j$$

$$\frac{ds_j}{dt} = Y_j \rho_j - D s_j$$

Where $\rho_j$ represents the reaction rate catalysed by the microbial proxy $m_j$. The next step of the approach consists of building an asymptotic observer that will enable us to estimate the dynamics of $s_j$ from measurements of $m_j$. For that, let us consider the following state transformation:

$$z_j = s_j - Y_j m_j$$

By deriving in time [5], we get

$$\frac{dz_j}{dt} = -D\Big[z_j + (1-\alpha)Y_j m_j\Big]$$

Let us denote $\hat{z}_j$ an on-line estimate of $z_j$. The dynamics of $\hat{z}_j$ follows

$$\frac{d\hat{z}_j}{dt} = -D\Big[\hat{z}_j + (1-\alpha)Y_j m_j\Big]$$

To simulate $\hat{z}_j$, we require dynamic data of $m_j$ and the parameters $\alpha$, $Y_j$ to be known. Under these conditions, we can use the estimate $\hat{z}_j$ and the dynamic data $m_j$ to provide an estimate $\hat{s}_j$ at each sampling time:

$$\hat{s}_j = \hat{z}_j + Y_j m_j$$

The asymptotic observer is the conjunction of the measurements $m_j$ with equations [7, 8]. As mentioned before, the approach does not require an explicit definition of the kinetic rate function $\rho_j$. We applied this development for acetate, butyrate and propionate, which requires measurements of the microbial proxies $m_{ac}$, $m_{bu}$, $m_{pr}$ and the yields $Y_{ac}$, $Y_{bu}$, $Y_{pr}$.

For both *in vitro* and *in vivo* case studies, the specific microbial proxy associated to the production of each VFA was derived from rumen microbial time series data determined by 16S rRNA gene amplicon sequencing. Raw data were analyzed using QIIME2 (version 2021.8) [37] using the script (run-samples) in https://github.com/frubino/cowpi and https://doi.org/10.5281/zenodo.8401851. Data were imported and denoised using the trim length option set at 400 and clustered using vsearch [38], in particular the 'cluster-features-de-novo'command with a percent identity set to 0.99. Classification was performed using the 'feature-classifier classify-consensus-blast'that uses BLAST [39] to classify the representative sequences, using default options and the Silva database (version 138) [40]. Finally, the OTU table representing microbial abundance time series was exported to be used in subsequent analysis steps. The resulting OTU table and representative sequences were then analyzed using CowPI [41] and modified using the information available at https://github.com/frubino/cowpi. The approach used here includes scaling data using DESeq2 [42] and infers modules from KEGG [43] instead of pathways, using the information in CowPi data. The pipeline for the inference of the functional modules is available at https://github.com/frubino/cowpi. KEGG modules resolve more specific processes and allow for greater precision when performing metabolic analysis, and like pathways, genes are their building blocks, so CowPI data can be used. For our analysis, modules allowed us to focus on more detailed aspects of metabolism for our model that pathways would not permit. Consequently, we used information about modules instead of pathways. However, there was no module in KEGG to present the butyrate metabolism. Therefore, an additional module for the metabolism of butyrate was introduced using information from [44]. These modules are the microbial proxies $m_j$. The abundances of the microbial proxies are relative measurements. However, since the data are scaled, we assumed that the absolute concentrations of microbial proxies are proportional to their relative abundances. As mentioned above, the asymptotic observer requires $\alpha$ and the yield factors to be known. The factor $\alpha$ was set to 0.85 according to [45]. We estimated the yield factors *via* the maximum likelihood (ML) approach as implemented in the Matlab IDEAS toolbox [46], which is freely available at http://genome.jouy.inra.fr/logiciels/IDEAS. For the optimization step, the Nelder–Mead Simplex method [47] implemented in the fminseach function was used. The model performance was

assessed by computing the coefficient of variation of the root mean squared error (CVRMSE). The Matlab scripts are available at [48] https://doi.org/10.5281/zenodo.8386786.

## Results

The scripts for the inference of the functional microbial modules are available at https://github.com/frubino/cowpi and https://doi.org/10.5281/zenodo.8401851. The abundances of each module for each case study are in the Tables modules.Rusitec.xls and modules.Cows in [48]. A total of 308 modules were identified. The microbial proxies for VFA production are named as M00579 ($m_{ac}$), M99999 ($m_{bu}$), M00013 ($m_{pr}$). The implementation of the observer for each case study is available at [48].

### *In vitro* case study

Table 1 shows the estimated abundance of the microbial proxies of VFA production. The values are the sum of the abundances from the liquid and solid phase. Fig 1 shows the dynamics of VFA (acetate, butyrate, and propionate) compared to the estimated VFA concentrations by the observer for the *in vitro* experiments with grass and hay using the rumen inoculum from two cows. Table 2 shows the model evaluation in terms of the CVRMSE. The model performance is satisfactory with average CVRMSE for acetate, butyrate and propionate of 9.8%, 14% and 14.5%. Table 3 shows the estimated yield factors for each diet and inoculum. The average values of the yields of acetate, butyrate and propionate are $0.13 \times 10^{-6}$, $0.39 \times 10^{-7}$ and $0.49 \times 10^{-6}$.

### *In vivo* case study

Table 4 shows the estimated abundance of the microbial proxies of VFA production from the liquid phase. Fig 2 shows the dynamics of VFA compared to the estimated VFA concentrations by the observer.

Table 5 shows the CVRMSE for the VFA concentrations. The model performance is adequate with average CVRMSE for acetate, butyrate and propionate of 16.4%, 15.8% and 19.8%. As shown in Table 6, the model performance is very satisfactory with regard to prediction of VFA molar proportions. The CVRMSE for the molar proportions of acetate, butyrate and

**Table 1. Microbial abundance time series of functional proxies of VFA production for the *in vitro* case study.** Values are the sum of the abundances from the liquid and solid phase.

| | | Sampling time (h) | | | |
|---|---|---|---|---|---|
| | | **2** | **4** | **8** | **24** |
| Rumen inoculum 1-Grass | $m_{ac}$ | 33139 | 37601 | 60680 | 70338 |
| | $m_{bu}$ | 52574 | 57719 | 116469 | 109282 |
| | $m_{pr}$ | 3503 | 3669 | 8181 | 8971 |
| Rumen inoculum 1-Hay | $m_{ac}$ | 14487 | 42303 | 36663 | 82646 |
| | $m_{bu}$ | 18600 | 61808 | 49934 | 118876 |
| | $m_{pr}$ | 640 | 4323 | 3673 | 13470 |
| Rumen inoculum 2-Grass | $m_{ac}$ | 87093 | 91845 | 105360 | 83174 |
| | $m_{bu}$ | 184833 | 206660 | 227768 | 235760 |
| | $m_{pr}$ | 23114 | 27944 | 24257 | 36697 |
| Rumen inoculum 2-Hay | $m_{ac}$ | 111166 | 106780 | 101168 | 102789 |
| | $m_{bu}$ | 240822 | 240218 | 194157 | 218833 |
| | $m_{pr}$ | 31287 | 31758 | 22945 | 22005 |

$m_{ac}$, $m_{bu}$, $m_{pr}$: abundances of microbial functional proxies for acetate, butyrate and propionate production.

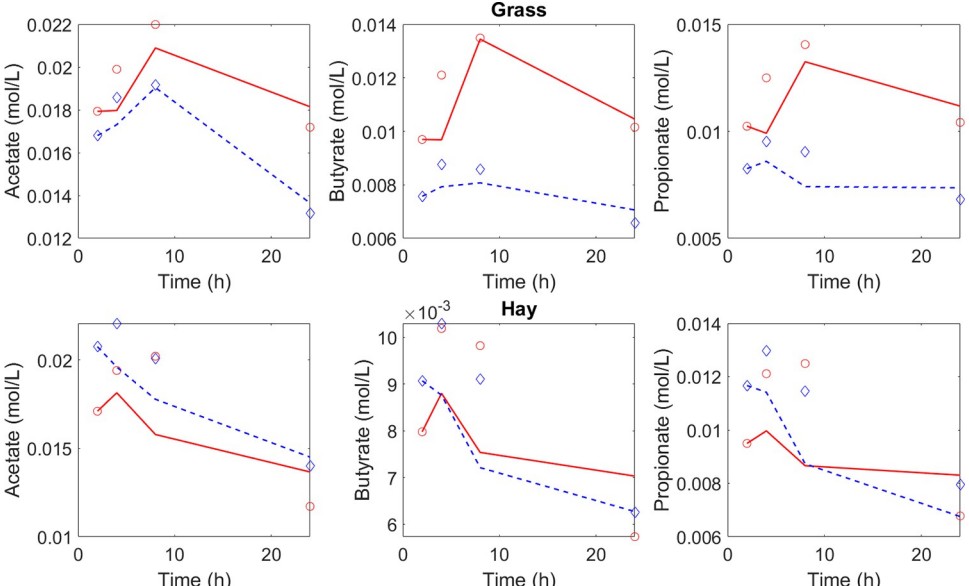

**Fig 1. Experimental data of VFA concentrations from a RUSITEC experiment with two feeds (grass and hay) and rumen fluid from two cows (○: cow 1; ◇: cow 2) are compared against the estimated concentrations by the state observer (red solid line for cow 1, dashed blue line for cow 2).**

propionate are 3.1%, 3.8% and 8.9%. Table 7 shows the estimated yield factors for each cow. The average values of the yields of acetate, butyrate and propionate are 1.43 x10$^{-6}$, 1.62x10$^{-7}$ and 2.43x10$^{-6}$.

## Discussion

The objective of this work was to integrate rumen microbiota time series determined by 16S rRNA gene amplicon sequencing into a mathematical model linking microbial data to the dynamics of the volatile fatty acids (VFA) production during rumen fermentation. This objective followed the rational that microbial data can be used to enhance predictive capabilities of rumen fermentation models. Our model development provided satisfactory results for estimating the dynamics of VFA from microbial data. The *in vivo* study showed that yield factors for the specific VFA production are similar between the cows. It will be useful to test our approach with animals fed at different diets including diets supplemented with additives impacting rumen fermentation patterns.

Given the limited number of animals used in our study, it is difficult to provide a fair comparison with existing rumen models. Furthermore, few rumen modelling studies have compared model predictions against dynamic data of VFA concentrations, with the exception of

**Table 2. Coefficient of variation of the root mean square error (%) for the observer applied to *in vitro* data of VFA concentrations.**

|  | Acetate | Butyrate | Propionate |
|---|---|---|---|
| Rumen inoculum 1- Grass | 7.1 | 11.8 | 13.2 |
| Rumen inoculum 2- Grass | 4.6 | 7.8 | 1.3 |
| Rumen inoculum 1- Hay | 16.9 | 20 | 25.6 |
| Rumen inoculum 2- Hay | 10.5 | 16.4 | 18 |
| Mean ± s.d. | 9.8 ± 5.3 | 14 ± 5.3 | 14.5 ± 10.2 |

**Table 3. Estimated yield parameters for the *in vitro* data.**

|  | $Y_{ac}$ | $Y_{bu}$ | $Y_{pr}$ |
|---|---|---|---|
| Rumen inoculum 1- Grass | $0.19 \times 10^{-6}$ | $0.77 \times 10^{-7}$ | $0.93 \times 10^{-6}$ |
| Rumen inoculum 2-Grass | $0.17 \times 10^{-6}$ | $0.26 \times 10^{-7}$ | $0.14 \times 10^{-6}$ |
| Rumen inoculum 1-Hay | $0.07 \times 10^{-6}$ | $0.30 \times 10^{-7}$ | $0.29 \times 10^{-6}$ |
| Rumen inoculum 2-Hay | $0.11 \times 10^{-6}$ | $0.22 \times 10^{-7}$ | $0.23 \times 10^{-6}$ |
| Mean ± s.d. | $0.13 \times 10^{-6} \pm 5.3 \times 10^{-8}$ | $0.39 \times 10^{-7} \pm 2.54 \times 10^{-8}$ | $0.40 \times 10^{-6} \pm 3.61 \times 10^{-7}$ |

$Y_{ac}, Y_{bu}, Y_{pr}$: yields of acetate, butyrate and propionate production (mol VFA / microbial functional proxy abundance).

the work of [16]. Under this context, however, it should be mentioned that the CVRMSE of our modelling approach are lower than those reported in [16], which strengthens our model development. From the modelling perspective, the structure of the model developed here is very simple compared to the structure of the existing rumen models. This parsimonious property is of usefulness for the development of *in silico* tools to predict rumen function from microbial data within a Precision Livestock Farming context. It should be said, however, that our study is a theoretical work that demonstrates the proof of concept that rumen microbial data can be used to predict rumen fermentation variables. However, the application of our modelling approach in farm conditions is currently unfeasible due to the need of rumen sampling and the costs of microbial analysis. To overcome the obstacle of rumen accessibility, it will be useful to test our approach using buccal microbial samples as proxies of the rumen microbiota [49, 50]. Buccal sampling might be a solution for getting quick samples and if it is coupled with easy sequencing instrument, it might become possible to use this approach to estimate rumen VFA using our modelling approach.

The fundamental aspect of our approach is the definition of the microbial proxies for each VFA. The microbial proxies here developed followed the principle that the rumen ecosystem operates as supra-organism provided with all the metabolic capabilities of its species. This approach does not account for species connectivity which is a relevant aspect in gut microbial ecosystems [51]. It is indeed interesting that the aggregated microbial proxies provided an

**Table 4. Microbial abundance time series of functional proxies of VFA production for the *in vivo* case study.** Values are from the liquid phase.

|  |  | Sampling time (h) | | | | | | | | | | | | | | |
|---|---|---|---|---|---|---|---|---|---|---|---|---|---|---|---|---|
|  |  | 7 | 10 | 13 | 16 | 19 | 22 | 25 | 28 | 31 | 34 | 37 | 40 | 43 | 46 | 49 | 52 |
| Cow 1 | $m_{ac}$ | 47708 | 45126 | 55602 | 57647 | 64097 | 56031 | 54965 | 65061 | 54497 | 69391 | 57204 | 67378 | 63766 | 67489 | 57861 | 56384 |
|  | $m_{bu}$ | 76650 | 72740 | 92161 | 95845 | 108538 | 89023 | 90047 | 108791 | 90950 | 117048 | 94074 | 115549 | 109147 | 112391 | 96751 | 97451 |
|  | $m_{pr}$ | 7229 | 6963 | 10052 | 9955 | 11753 | 8104 | 8651 | 11608 | 9396 | 12755 | 10278 | 13197 | 11730 | 11249 | 10067 | 11329 |
| Cow 2 | $m_{ac}$ | 49745 | 48073 | 52536 | 49837 | 59831 | 58632 | 58376 | 40248 | 66988 | 69821 | 67276 | 54960 | 64366 | 50587 | 58103 | 54381 |
|  | $m_{bu}$ | 80804 | 80864 | 87082 | 84439 | 98867 | 102169 | 97994 | 67514 | 113688 | 118291 | 114353 | 94216 | 110819 | 87848 | 100618 | 96853 |
|  | $m_{pr}$ | 8064 | 8626 | 9009 | 8991 | 9808 | 11502 | 9774 | 6934 | 12201 | 12928 | 12633 | 10080 | 12292 | 9732 | 11186 | 10745 |
| Cow 3 | $m_{ac}$ | 42144 | 48535 | 50274 | 55855 | 45496 | 58116 | 49365 | 54277 | 52852 | 66731 | 60735 | 53184 | 54231 | 65121 | 49727 | 51256 |
|  | $m_{bu}$ | 73426 | 84226 | 85419 | 97337 | 78949 | 102467 | 80924 | 93929 | 89409 | 111399 | 102816 | 90607 | 92020 | 110914 | 84625 | 91281 |
|  | $m_{pr}$ | 8149 | 9554 | 9264 | 11066 | 8440 | 11366 | 7839 | 10324 | 9596 | 11580 | 11227 | 10341 | 9335 | 11322 | 9163 | 10974 |
| Cow 4 | $m_{ac}$ | 46298 | 12944 | 58665 | 60244 | 54556 | 55437 | 53430 | 54539 | 60154 | 64751 | 56813 | 56088 | 58200 | 53288 | 52679 | 57640 |
|  | $m_{bu}$ | 77495 | 24482 | 99615 | 100443 | 90413 | 95868 | 86833 | 90640 | 103506 | 114801 | 93832 | 93850 | 101479 | 89818 | 91936 | 98357 |
|  | $m_{pr}$ | 8128 | 2847 | 10811 | 10838 | 9539 | 10574 | 8927 | 9399 | 12079 | 13578 | 10265 | 9756 | 11216 | 9414 | 10771 | 11167 |

$m_{ac}, m_{bu}, m_{pr}$: abundances of microbial functional proxies for acetate, butyrate and propionate production.

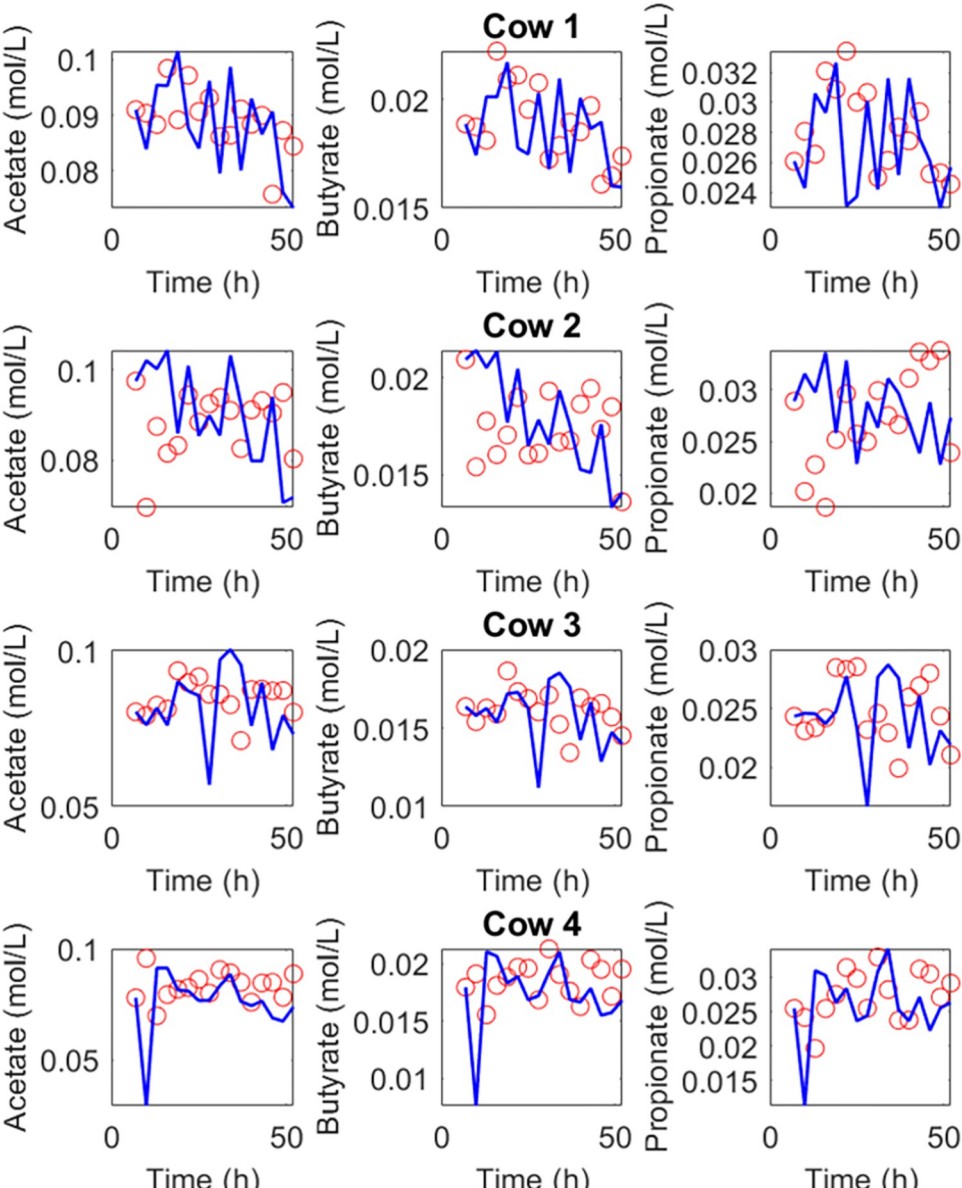

**Fig 2. Experimental data of VFA concentrations (○) from an *in vivo* experiment with four cows are compared against the estimated concentrations by the state observer (blue solid line).**

**Table 5. Coefficient of variation of the root mean square error (%) for the observer applied to *in vivo* data of VFA concentrations.**

|  | Acetate | Butyrate | Propionate |
|---|---|---|---|
| Cow 1 | 10.1 | 10.2 | 14.7 |
| Cow 2 | 15.4 | 14.1 | 17.1 |
| Cow 3 | 16.5 | 18.4 | 25.9 |
| Cow 4 | 23.8 | 20.5 | 21.3 |
| Mean± s.d. | 16.4 ± 5.6 | 15.8 ± 4.6 | 19.8 ± 4.9 |

**Table 6. Coefficient of variation of the root mean square error (%) for the observer applied to *in vivo* data of VFA molar proportions.**

|  | Acetate | Butyrate | Propionate |
|---|---|---|---|
| Cow 1 | 2.9 | 3.4 | 8.3 |
| Cow 2 | 1.8 | 2.8 | 5.7 |
| Cow 3 | 3.6 | 4.6 | 10.9 |
| Cow 4 | 4.2 | 4.5 | 10.5 |
| Mean± s.d. | 3.1 ± 1.0 | 3.8 ± 0.9 | 8.9 ± 2.4 |

**Table 7. Estimated yield parameters for the *in vivo* data.**

|  | $Y_{ac}$ | $Y_{bu}$ | $Y_{pr}$ |
|---|---|---|---|
| Cow 1 | $1.38 \times 10^{-6}$ | $1.73 \times 10^{-7}$ | $2.37 \times 10^{-6}$ |
| Cow 2 | $1.52 \times 10^{-6}$ | $1.55 \times 10^{-7}$ | $2.16 \times 10^{-6}$ |
| Cow 3 | $1.43 \times 10^{-6}$ | $1.50 \times 10^{-7}$ | $2.64 \times 10^{-6}$ |
| Cow 4 | $1.39 \times 10^{-6}$ | $1.83 \times 10^{-7}$ | $2.54 \times 10^{-6}$ |
| Mean ± s.d. | $1.43 \times 10^{-6} \pm 6.28 \times 10^{-8}$ | $1.62 \times 10^{-7} \pm 1.56 \times 10^{-8}$ | $2.43 \times 10^{-6} \pm 2.10 \times 10^{-7}$ |

$Y_{ac}$, $Y_{bu}$, $Y_{pr}$: yields of acetate, butyrate and propionate production (mol VFA / microbial functional proxy abundance).

adequate representation of VFA dynamics. Although the results are promising, future research is needed to refine the definition of these proxies, by taking into account quantitative transcriptomic information to identify the actual metabolic activity.

Our study has taken a step in the direction of capitalizing on microbial data to predict rumen function. In this first step, we used asymptotic state observers due to its simple structure and the advantage of not requiring information on the mathematical functions describing the metabolic kinetic rates. However, asymptotic observers have the limitation that the convergence rate cannot be tuned because it is determined by operational conditions (*e.g.*, ruminal fractional passage rate). This limitation can be overcome by other observers. However, more sophisticated technicalities are needed for the design and implementation of such observers [26]. Our approach was applied to estimate VFA dynamics. We think that the approach can be extended to estimate other metabolites including methane production by the rumen methanogenic archaea. Such an application will be of great interesting for the monitoring of methane emissions and for the evaluation of methane inhibition strategies.

## Conclusions

Existing mechanistic models of rumen fermentation consider an aggregated representation of the rumen microbiota and its metabolic function. However, none of these models integrate microbial genomic knowledge and thus do not capitalize on the rich information of microbial genomic sequencing. In this work, we integrated microbial time series of the rumen microbiota determined by 16S rDNA into a dynamic model of the rumen microbiome. Our results showed the estimated VFA concentrations converge towards the real VFA concentration dynamics demonstrating the promising potential of our approach.

## Acknowledgments

Rafael Muñoz-Tamayo thanks Pablo Ugalde-Salas (Inria, France) and Jerôme Harmand (INRAE, France) for their helpful explanations on the use of state observers for microbial

ecosystems. The Finnish Functional Genomics Centre supported by the University of Turku, Åbo Akademi University, and Biocenter Finland is acknowledged for sequencing. The authors wish to acknowledge CSC–IT Center for Science, Finland, for computational resources.

## Author Contributions

**Conceptualization:** Mohsen Davoudkhani, Francesco Rubino, Christopher J. Creevey, Seppo Ahvenjärvi, Ali R. Bayat, Ilma Tapio, Alejandro Belanche, Rafael Muñoz-Tamayo.

**Data curation:** Francesco Rubino, Ilma Tapio, Alejandro Belanche.

**Formal analysis:** Mohsen Davoudkhani, Francesco Rubino, Rafael Muñoz-Tamayo.

**Investigation:** Seppo Ahvenjärvi, Ali R. Bayat, Ilma Tapio, Alejandro Belanche.

**Methodology:** Mohsen Davoudkhani, Francesco Rubino, Christopher J. Creevey, Rafael Muñoz-Tamayo.

**Project administration:** Rafael Muñoz-Tamayo.

**Software:** Mohsen Davoudkhani, Francesco Rubino, Rafael Muñoz-Tamayo.

**Supervision:** Christopher J. Creevey, Rafael Muñoz-Tamayo.

**Writing – original draft:** Mohsen Davoudkhani, Francesco Rubino, Rafael Muñoz-Tamayo.

**Writing – review & editing:** Christopher J. Creevey, Seppo Ahvenjärvi, Ali R. Bayat, Ilma Tapio, Alejandro Belanche.

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
