## [Decision Letter · Decision Letter 0]

20 Dec 2023

PONE-D-23-32553Integrating microbial abundance time series with fermentation dynamics of the rumen microbiome via mathematical modellingPLOS ONE

Dear Dr. Muñoz-Tamayo,

Thank you for submitting your manuscript to PLOS ONE. After careful consideration, we feel that it has merit but does not fully meet PLOS ONE’s publication criteria as it currently stands. Therefore, we invite you to submit a revised version of the manuscript that addresses the points raised during the review process.

We look forward to receiving your revised manuscript.

Kind regards,

Adham A. Al-Sagheer

Academic Editor

PLOS ONE

Integration of microbial time series into a mechanistic model of the rumen microbiome under the Rusitec condition - https://doi.org/10.1016/j.anscip.2022.07.443

In your revision ensure you cite all your sources (including your own works), and quote or rephrase any duplicated text outside the methods section. Further consideration is dependent on these concerns being addressed.

“All authors receive funding from the MASTER project, an Innovation Action funded by the European Union’s Horizon 2020 research and innovation programme under grant agreement No 818368.”

4. We noted in your submission details that a portion of your manuscript may have been presented or published elsewhere. [Belanche, A., Newbold, C.J., Lin, W., Rees Stevens, P., Kingston-Smith, A.H., 2017. A systems biology approach reveals differences in the dynamics of colonization and degradation of grass vs. hay by rumen microbes with minor effects of vitamin E supplementation. Front. Microbiol. 8. https://doi.org/10.3389/fmicb.2017.01456]. Please clarify whether this [conference proceeding or publication] was peer-reviewed and formally published. If this work was previously peer-reviewed and published, in the cover letter please provide the reason that this work does not constitute dual publication and should be included in the current manuscript.

Reviewers' comments:

Reviewer's Responses to Questions

**Comments to the Author**

1. Is the manuscript technically sound, and do the data support the conclusions?

Reviewer #1: Yes

Reviewer #2: No

2. Has the statistical analysis been performed appropriately and rigorously? 

Reviewer #1: Yes

Reviewer #2: Yes

3. Have the authors made all data underlying the findings in their manuscript fully available?

Reviewer #1: Yes

Reviewer #2: Yes

4. Is the manuscript presented in an intelligible fashion and written in standard English?

Reviewer #1: Yes

Reviewer #2: Yes

5. Review Comments to the Author

**Reviewer #1: **

In my opinion, This manuscript aims to a correlation between microbial data and the dynamics of volatile fatty acids (VFA) production during fermentation. The present study employed state observer theory to develop a model, which was validated using data from an in vitro RUSITEC experiment and an in vivo experiment involving four cows. Overall, the manuscript is coherent and clear. Good data presentation has been shown in your manuscript.The manuscript can be accepted after a few modifications as follows.

This manuscript aims to a correlation between microbial data and the dynamics of volatile fatty acids (VFA) production during fermentation. The present study employed state observer theory to develop a model, which was validated using data from an in vitro RUSITEC experiment and an in vivo experiment involving four cows. Overall, the manuscript is coherent and clear. Good data presentation has been shown in your manuscript. I make some comments to be considered below:

As mentioned in line 104,” here we propose a novel approach that uses OTU data to derive microbial functional proxies for specific process of rumen metabolism such as volatile fatty acid (VFA)” Can you explain the content of OTU data in more detail?
As mentioned in line 118-120，“Fermentation vessels (800 mL effective volume) were inoculated with rumen fluid and incubated with the  experimental diets (80:20 forage to concentrate ratio).” Why was only a single forage concentrate ratio of 80:20 selected for the experimental diet? Shouldn't multiple ratios have been considered?
In Tables 1 to 7, the terms Cow1 Grass, Cow1, and Cow1 Hay have been frequently observed. Although the term Cow1 Grass has been used to differentiate them from Cow1 originating from different experiments, the abundance of these occurrences can still lead to confusion among data objects. Therefore, it is necessary to explore alternative methods for distinguishing between these two groups of subjects.
Line 288-289, which section does this paragraph belong to? If it is a description of the contents in the table 3, please check the format requirements.
The thickness of the lines in Fig 1 and Fig 2 are inconsistent. The chart presentation in a paper should adhere to standardized formatting.

**Reviewer #2: **

Dear Editor

Thank you for giving me the opportunity to evaluate this manuscript entitled “Integrating microbial abundance time series with fermentation dynamics of the rumen microbiome via mathematical modelling”. The manuscript was properly conducted and findings reported are important for PLOS ONE. The revised paper contains valuable data. The authors investigated an interesting topic and the objective of the paper is of worldwide interest and fits well within the overall scope of the journal. Results were properly reported and the findings have been accurately discussed and compared with other published papers. So, based on my opinion the manuscript merits acceptance.

Regards

6. PLOS authors have the option to publish the peer review history of their article (what does this mean?). If published, this will include your full peer review and any attached files.

Reviewer #1: No

Reviewer #2: **Yes: **

---

## [Author Response · Author response to Decision Letter 0]

21 Dec 2023

The responses are given in the response letter

---

## [Editor Report · Decision Letter 1]

2 Feb 2024

Integrating microbial abundance time series with fermentation dynamics of the rumen microbiome via mathematical modelling

PONE-D-23-32553R1

Dear Dr. Muñoz-Tamayo,

We’re pleased to inform you that your manuscript has been judged scientifically suitable for publication and will be formally accepted for publication once it meets all outstanding technical requirements.

Kind regards,

Adham A. Al-Sagheer

Academic Editor

PLOS ONE
---

## [Editor Report · Acceptance letter]

11 Mar 2024

PONE-D-23-32553R1 

PLOS ONE

Dear Dr. Muñoz-Tamayo, 

I'm pleased to inform you that your manuscript has been deemed suitable for publication in PLOS ONE. Congratulations! Your manuscript is now being handed over to our production team.

Kind regards, 

on behalf of

Dr. Adham A. Al-Sagheer 

Academic Editor

PLOS ONE